# Urban–rural disparity in cancer incidence in China, 2008–2012: a cross-sectional analysis of data from 36 cancer registers

Shuai Yuan [ORCID],[1] Shao-Hua Xie[2]

¹Institute of Environmental Medicine, Karolinska Institutet, Stockholm, Sweden
²Upper Gastrointestinal Surgery, Department of Molecular Medicine and Surgery, Karolinska Institutet, Karolinska University Hospital, Stockholm, Sweden

**Correspondence to**
Dr Shao-Hua Xie;
shaohua.xie@ki.se

## ABSTRACT

**Objective** The substantial differences in socioeconomic and lifestyle exposures between urban and rural areas in China may lead to urban–rural disparity in cancer risk. This study aimed to assess the urban–rural disparity in cancer incidence in China.

**Methods** Using data from 36 regional cancer registries in China in 2008–2012, we compared the age-standardised incidence rates of cancer by sex and anatomic site between rural and urban areas. We calculated the rate difference and rate ratio comparing rates in rural versus urban areas by sex and cancer type.

**Results** The incidence rate of all cancers in women was slightly lower in rural areas than in urban areas, but the total cancer rate in men was higher in rural areas than in urban areas. The incidence rates in women were higher in rural areas than in urban areas for cancers of the oesophagus, stomach, and liver and biliary passages, but lower for cancers of thyroid and breast. Men residing in rural areas had higher incidence rates for cancers of the oesophagus, stomach, and liver and biliary passages, but lower rates for prostate cancer, lip, oral cavity and pharynx cancer, and colorectal cancer.

**Conclusions** Our findings suggest substantial urban–rural disparity in cancer incidence in China, which varies across cancer types and the sexes. Cancer prevention strategies should be tailored for common cancers in rural and urban areas.

## Strengths and limitations of this study

► We used data from the 36 cancer registers of high quality to investigate urban–rural disparity for overall and site-specific cancer incidence by sex and on both absolute and relative scales.

► The cross-sectional design limited the causal inference in geographical disparities to cancer incidence.

► Included registries only represented a limited proportion of the total population in China and were not in well balanced distribution regarding geographical regions.

► We were not able to analyse the data by histological type.

► We could not evaluate the associations of cancer disparity with gross domestic product, human development index or corresponding risk factors.

## INTRODUCTION

Cancer is one of the leading causes of death worldwide, including in China. According to China Health Statistics Yearbook, an estimated 2.2 million cancer deaths occurred in China in 2017, accounting for around 25% of all deaths in this year.[1] Moreover, the past few decades have witnessed rapid urbanisation and industrialisation in China, and the uneven distribution of wealth and lifestyle profiles across regions in the nation during this process may have in turn resulted in substantial urban–rural disparity in cancer risk. Previous studies have suggested marked urban–rural disparity in cancer incidence in China, and such disparity seems varying across cancer types.[2–8] However, most of these previous studies were limited in a certain area or province, or specific types of cancer only. Due to historically limited population-based cancer registry data in China, the urban–rural disparity in cancer incidence has not been well characterised.

China has recently established the National Central Cancer Registry and data from 36 regional cancer registries have been included in the latest release of *Cancer Incidence in Five Continents* series published by the International Agency for Research on Cancer.[9] This provides a new opportunity to assess the regional disparity in cancer incidence in China on a national scale. Therefore, using data from these high-quality registries, we conducted the present study to characterise the urban–rural disparity in total and type-specific cancer incidence by sex in China in 2008–2012.

## METHODS
### Data source

We extracted data on cancer incidence and population sizes from 36 regional cancer registries, including 14 registries in rural areas

and 22 in urban areas, in China during 2008–2012 which are included in the XI volume of *Cancer Incidence in Five Continents* series published by the International Agency for Research on Cancer.[10] These registries are located in different geographical regions throughout the country. Cancer cases were defined by International Classification of Disease codes, which are listed in a webpage of the International Agency for Research on Cancer (https://ci5.iarc.fr/CI5-XI/PDF/Chapter3.pdf). Detailed information on the included registries is presented in online supplemental tables 1 and 2. The case-weighted means of percentage of case defined by death certificate only were ~1.9 in men,~1.5 in women and ~1.7 in both men and women among included registries. Included population from 36 cancer registries made up ~5% of the total population in China. We pooled the numbers of cases and population sizes at risk from multiple registers separately for urban and rural areas by sex and cancer type (anatomic site).

## Statistical analysis

We calculated the crude and age-standardised incidence rates (ASRs) and their 95% CIs by sex and cancer type separately for rural and urban areas. The ASRs were computed by using the direct standardisation method with World Standard Population 2000 as reference.[11] The 95% CIs of crude rates were estimated under the assumption of Poisson distribution and CIs for ASRs were calculated based on the gamma distribution assuming that the standardised rate is a weighted sum of independent Poisson random variables.[12] The urban–rural disparities in cancer incidence were quantitatively assessed with two disparity measures (ie, rate difference (RD) on the absolute scale and rate ratio (RR) on the relative scale).[13] We calculated the RD (ASR in rural areas—ASR in urban areas) and RR (ratio of ASRs in rural areas relative to urban areas) with 95% CIs by sex and cancer type. All statistical analyses were performed using the SAS V.9.4 (SAS Institute, Cary, North Carolina, USA) and two-sided.

## Patient and public involvement

Patients or the public were not involved in the design, or conduct, or reporting, or dissemination plans of our research.

## Ethical considerations

The analyses were solely based on publicly available data of population sizes and aggregated number of cancer cases and as such, ethics approval or consent to participate was not deemed to be necessary. No individual-level data were involved in the study or in defining the research question or outcome measures.

## RESULTS
## Cancer incidence in rural areas

The crude rates and ASRs in rural and urban areas are displayed in table 1 for women and in table 2 for men. In

total, 58 576 female and 81 709 male new cancer cases were recorded in 14 cancer registers in rural areas in China during the period 2008–2012, contributing to the ASRs of 188.6 (95% CI 187.1 to 190.2) per 100 000 population per year in women and 273.4 (95% CI 271.5 to 275.2) per 100 000 population per year in men. In women residing in rural areas, the ASRs were observed for stomach cancer, followed by cancers of the oesophagus, lung, breast, and liver and biliary passages. In men in rural areas, stomach cancer was also the most common cancer type, followed by cancers of the lung, liver and biliary passages, oesophagus, and colon, rectum and anus.

## Cancer incidence in urban areas

A total of 388 917 women and 449 934 men were diagnosed with any cancer as recorded in the 22 registers in urban areas in China during the period 2008–2012. The ASRs of all cancers were 193.4 (95% CI 192.7 to 194.0) per 100 000 population per year in women and 233.4 (95% CI 232.7 to 234.1) per 100 000 population per year in men. The five most frequent cancers in women in urban areas were cancers of the breast, lung, colon and rectum, uterus, and thyroid. In men, the highest ASRs were observed for lung cancer, followed by cancer of colon and rectum, liver and biliary passages, stomach, and prostate.

## RD comparing rural and urban areas

Figure 1 presents the absolute difference in cancer incidence, that is, RDs comparing rural and urban areas, in women and in men. The total cancer incidence was lower in women (RD, −4.7 per 100 000 population per year, 95% CI −6.4 to −3.1) but higher in men (RD, 39.9, 95% CI 39.0 to 43.0) living in the rural areas than those in urban areas. Compared with those in urban areas, women living in rural areas had substantially higher incidence rates of cancers of the oesophagus, stomach, and liver and biliary passages cancer, but lower incidence rates of cancers of the breast, thyroid, colon and rectum, ovary, and lung. Men residing in the rural areas had higher incidence rates of cancers of the stomach, oesophagus, and liver and biliary passages, while those in urban areas had higher rates of cancers of colon and rectum, prostate cancer, and lip, oral cavity and pharynx, kidney, thyroid, and bladder.

## RR comparing rural and urban areas

The RRs measuring the urban–rural disparity cancer incidence on the relative scale are shown in figure 2. The total cancer incidence rates were similar between women residing in rural areas and those in urban areas as measured by RR (0.98, 95% CI 0.97 to 0.98). However, the total cancer incidence rate in men in rural areas was around 20% higher than those in urban areas (1.17, 95% CI 1.16 to 1.18). When analysed by cancer type, compared those in urban areas, women in rural areas had substantially higher incidence rates for cancers of the oesophagus and stomach, but at least 50% lower incidence rates for cancers of urinary organs other than kidney and bladder, thyroid cancer, kidney cancer, Hodgkin lymphoma, and

**Table 1** Cancer incidence rates by anatomic site in women in China, 2008–2012

| Cancer type | Rural areas | | | Urban areas | | |
|---|---|---|---|---|---|---|
| | Number of cases | Crude rate (95% CI) | Age-standardised rate (95% CI) | Number of cases | Crude rate (95% CI) | Age-standardised rate (95% CI) |
| Lip, oral cavity and pharynx | 850 | 3.7 (3.4 to 3.9) | 2.8 (2.6 to 3.0) | 8329 | 6.2 (6.1 to 6.3) | 4.3 (4.2 to 4.4) |
| Oesophagus | 8408 | 36.2 (35.4 to 36.9) | 26.3 (25.8 to 26.9) | 6034 | 4.5 (4.4 to 4.6) | 2.7 (2.6 to 2.8) |
| Stomach | 8410 | 36.2 (35.4 to 36.9) | 26.5 (25.9 to 27.1) | 22267 | 16.6 (16.4 to 16.8) | 10.5 (10.4 to 10.7) |
| Small intestine | 162 | 0.7 (0.6 to 0.8) | 0.5 (0.4 to 0.6) | 1818 | 1.4 (1.3 to 1.4) | 0.9 (0.8 to 0.9) |
| Colon, rectum and anus | 4268 | 18.4 (17.8 to 18.9) | 13.5 (13.1 to 13.9) | 47907 | 35.7 (35.4 to 36.1) | 22.3 (22.1 to 22.5) |
| Liver and biliary passages | 6084 | 26.2 (25.5 to 26.8) | 19.2 (18.8 to 19.7) | 24268 | 18.1 (17.9 to 18.3) | 11.2 (11.1 to 11.3) |
| Pancreas | 1555 | 6.7 (6.4 to 7.0) | 4.8 (4.6 to 5.0) | 10067 | 7.5 (7.4 to 7.7) | 4.6 (4.5 to 4.6) |
| Larynx | 81 | 0.3 (0.3 to 0.4) | 0.3 (0.2 to 0.3) | 508 | 0.4 (0.3 to 0.4) | 0.2 (0.2 to 0.3) |
| Lung | 7208 | 31.0 (30.3 to 31.7) | 22.6 (22.0 to 23.1) | 54148 | 40.4 (40.1 to 40.7) | 25.0 (24.8 to 25.2) |
| Bone | 448 | 1.9 (1.8 to 2.1) | 1.6 (1.4 to 1.7) | 1665 | 1.2 (1.2 to 1.3) | 1.0 (0.9 to 1.0) |
| Melanoma of skin | 109 | 0.5 (0.4 to 0.6) | 0.4 (0.3 to 0.4) | 905 | 0.7 (0.6 to 0.7) | 0.4 (0.4 to 0.5) |
| Skin, excluding melanoma | 563 | 2.4 (2.2 to 2.6) | 1.7 (1.6 to 1.9) | 5353 | 4.0 (3.9 to 4.1) | 2.4 (2.4 to 2.5) |
| Mesothelioma | 31 | 0.1 (0.1 to 0.2) | 0.1 (0.1 to 0.1) | 319 | 0.2 (0.2 to 0.3) | 0.2 (0.1 to 0.2) |
| Breast | 6539 | 28.1 (27.4 to 28.8) | 21.5 (21.0 to 22.0) | 80535 | 60.1 (59.7 to 60.5) | 41.3 (41 to 41.6) |
| Uterus | 5572 | 24 (23.3 to 24.6) | 18.4 (17.9 to 18.8) | 34763 | 25.9 (25.7 to 26.2) | 18.1 (17.9 to 18.3) |
| Ovary | 1269 | 5.5 (5.2 to 5.8) | 4.2 (4.0 to 4.5) | 12782 | 9.5 (9.4 to 9.7) | 6.7 (6.6 to 6.8) |
| Other female genital organs | 200 | 0.9 (0.7 to 1.0) | 0.7 (0.6 to 0.7) | 2038 | 1.5 (1.5 to 1.6) | 1.0 (1.0 to 1.1) |
| Kidney | 380 | 1.6 (1.5 to 1.8) | 1.3 (1.2 to 1.4) | 5642 | 4.2 (4.1 to 4.3) | 2.8 (2.8 to 2.9) |
| Bladder | 507 | 2.2 (2.0 to 2.4) | 1.6 (1.5 to 1.7) | 5029 | 3.8 (3.6 to 3.9) | 2.3 (2.2 to 2.3) |
| Other urinary organs | 73 | 0.3 (0.2 to 0.4) | 0.2 (0.2 to 0.3) | 2271 | 1.7 (1.6 to 1.8) | 1.0 (1.0 to 1.1) |
| Eye | 51 | 0.2 (0.2 to 0.3) | 0.2 (0.1 to 0.3) | 217 | 0.2 (0.1 to 0.2) | 0.2 (0.2 to 0.2) |
| Brain and nervous system | 1405 | 6.0 (5.7 to 6.4) | 4.9 (4.6 to 5.1) | 7457 | 5.6 (5.4 to 5.7) | 4.2 (4.1 to 4.3) |
| Thyroid | 999 | 4.3 (4.0 to 4.6) | 3.4 (3.2 to 3.7) | 22869 | 17.1 (16.8 to 17.3) | 13.0 (12.8 to 13.2) |
| Other endocrine organs | 165 | 0.7 (0.6 to 0.8) | 0.6 (0.5 to 0.7) | 1229 | 0.9 (0.9 to 1.0) | 0.7 (0.7 to 0.8) |
| Hodgkin lymphoma | 42 | 0.2 (0.1 to 0.2) | 0.2 (0.1 to 0.2) | 508 | 0.4 (0.3 to 0.4) | 0.3 (0.3 to 0.4) |
| Non-Hodgkin's lymphoma | 862 | 3.7 (3.5 to 4.0) | 2.9 (2.7 to 3.1) | 7672 | 5.7 (5.6 to 5.9) | 3.9 (3.8 to 4.0) |
| Multiple myeloma | 185 | 0.8 (0.7 to 0.9) | 0.6 (0.5 to 0.7) | 2162 | 1.6 (1.5 to 1.7) | 1.0 (1.0 to 1.1) |
| Lymphoid leukaemia | 1189 | 5.1 (4.8 to 5.4) | 4.6 (4.3 to 4.8) | 6565 | 4.9 (4.8 to 5.0) | 4.2 (4.1 to 4.3) |
| Other and unspecified | 961 | 4.1 (3.9 to 4.4) | 3.2 (3.0 to 3.4) | 13590 | 10.1 (10.0 to 10.3) | 6.7 (6.6 to 6.8) |
| All sites but skin | 58013 | 249.5 (247.5 to 251.5) | 186.9 (185.4 to 188.4) | 383564 | 286.2 (285.3 to 287.1) | 190.9 (190.3 to 191.5) |
| All sites | 58576 | 251.9 (249.9 to 253.9) | 188.6 (187.1 to 190.2) | 388917 | 290.2 (289.3 to 291.1) | 193.4 (192.7 to 194) |

**Table 2** Cancer incidence rate by anatomic site in men in China, 2008–2012

| Cancer type | Rural areas | | | Urban areas | | |
|---|---|---|---|---|---|---|
| | Number of cases | Crude rate (95% CI) | Age-standardised rate (95% CI) | Number of cases | Crude rate (95% CI) | Age-standardised rate (95% CI) |
| Lip, oral cavity and pharynx | 1517 | 6.3 (6.0 to 6.6) | 5.0 (4.8 to 5.3) | 18962 | 14.2 (14.0 to 14.4) | 10.1 (10.0 to 10.2) |
| Oesophagus | 13139 | 54.5 (53.6 to 55.4) | 43.5 (42.7 to 44.2) | 21961 | 16.5 (16.2 to 16.7) | 11.1 (11.0 to 11.3) |
| Stomach | 18515 | 76.8 (75.7 to 77.9) | 61.3 (60.4 to 62.2) | 45179 | 33.9 (33.5 to 34.2) | 23.0 (22.8 to 23.2) |
| Small intestine | 190 | 0.8 (0.7 to 0.9) | 0.6 (0.5 to 0.7) | 2238 | 1.7 (1.6 to 1.7) | 1.2 (1.1 to 1.2) |
| Colon, rectum and anus | 5121 | 21.2 (20.7 to 21.8) | 17.1 (16.6 to 17.6) | 60205 | 45.1 (44.8 to 45.5) | 30.7 (30.4 to 30.9) |
| Liver and biliary passages | 13885 | 57.6 (56.7 to 58.6) | 45.8 (45.1 to 46.6) | 56100 | 42.0 (41.7 to 42.4) | 29.0 (28.7 to 29.2) |
| Pancreas | 1802 | 7.5 (7.1 to 7.8) | 6.0 (5.8 to 6.3) | 12593 | 9.4 (9.3 to 9.6) | 6.4 (6.3 to 6.5) |
| Larynx | 522 | 2.2 (2.0 to 2.4) | 1.7 (1.6 to 1.9) | 5679 | 4.3 (4.1 to 4.4) | 2.9 (2.8 to 3.0) |
| Lung | 15740 | 65.3 (64.3 to 66.3) | 52.7 (51.9 to 53.5) | 104382 | 78.2 (77.7 to 78.7) | 52.8 (52.5 to 53.1) |
| Bone | 612 | 2.5 (2.3 to 2.7) | 2.2 (2.0 to 2.4) | 2041 | 1.5 (1.5 to 1.6) | 1.2 (1.2 to 1.3) |
| Melanoma of skin | 119 | 0.5 (0.4 to 0.6) | 0.4 (0.3 to 0.5) | 922 | 0.7 (0.6 to 0.7) | 0.5 (0.5 to 0.5) |
| Skin, excluding melanoma | 542 | 2.2 (2.1 to 2.4) | 1.9 (1.7 to 2.0) | 5611 | 4.2 (4.1 to 4.3) | 2.9 (2.8 to 3.0) |
| Mesothelioma | 30 | 0.1 (0.1 to 0.2) | 0.1 (0.1 to 0.1) | 477 | 0.4 (0.3 to 0.4) | 0.2 (0.2 to 0.3) |
| Breast | 68 | 0.3 (0.2 to 0.4) | 0.2 (0.2 to 0.3) | 677 | 0.5 (0.5 to 0.5) | 0.4 (0.3 to 0.4) |
| Prostate | 1152 | 4.8 (4.5 to 5.1) | 4.0 (3.8 to 4.3) | 25963 | 19.5 (19.2 to 19.7) | 13.0 (12.8 to 13.2) |
| Testis | 103 | 0.4 (0.3 to 0.5) | 0.4 (0.3 to 0.5) | 1176 | 0.9 (0.8 to 0.9) | 0.8 (0.8 to 0.9) |
| Other male genital organs | 184 | 0.8 (0.7 to 0.9) | 0.6 (0.5 to 0.7) | 1371 | 1.0 (1.0 to 1.1) | 0.7 (0.7 to 0.8) |
| Kidney | 535 | 2.2 (2.0 to 2.4) | 1.8 (1.7 to 2.0) | 11201 | 8.4 (8.2 to 8.6) | 5.9 (5.8 to 6.0) |
| Bladder | 1610 | 6.7 (6.4 to 7.0) | 5.5 (5.2 to 5.7) | 16013 | 12.0 (11.8 to 12.2) | 8.1 (8.0 to 8.2) |
| Other urinary organs | 121 | 0.5 (0.4 to 0.6) | 0.4 (0.3 to 0.5) | 2449 | 1.8 (1.8 to 1.9) | 1.2 (1.2 to 1.3) |
| Eye | 49 | 0.2 (0.2 to 0.3) | 0.2 (0.1 to 0.3) | 236 | 0.2 (0.2 to 0.2) | 0.2 (0.2 to 0.2) |
| Brain and nervous system | 1496 | 6.2 (5.9 to 6.5) | 5.3 (5.0 to 5.5) | 7454 | 5.6 (5.5 to 5.7) | 4.4 (4.3 to 4.5) |
| Thyroid | 293 | 1.2 (1.1 to 1.4) | 1.0 (0.9 to 1.1) | 7193 | 5.4 (5.3 to 5.5) | 4.2 (4.1 to 4.3) |
| Other endocrine organs | 130 | 0.5 (0.5 to 0.6) | 0.5 (0.4 to 0.5) | 1221 | 0.9 (0.9 to 1.0) | 0.8 (0.7 to 0.8) |
| Hodgkin lymphoma | 63 | 0.3 (0.2 to 0.3) | 0.2 (0.2 to 0.3) | 740 | 0.6 (0.5 to 0.6) | 0.5 (0.4 to 0.5) |
| Non-Hodgkin's lymphoma | 1266 | 5.3 (5.0 to 5.5) | 4.3 (4.1 to 4.6) | 10025 | 7.5 (7.4 to 7.7) | 5.5 (5.4 to 5.6) |
| Multiple myeloma | 286 | 1.2 (1.1 to 1.3) | 0.9 (0.8 to 1.1) | 3096 | 2.3 (2.2 to 2.4) | 1.6 (1.5 to 1.6) |
| Lymphoid leukaemia | 1433 | 5.9 (5.6 to 6.3) | 5.4 (5.1 to 5.7) | 8751 | 6.6 (6.4 to 6.7) | 5.8 (5.6 to 5.9) |
| Other and unspecified | 1186 | 4.9 (4.6 to 5.2) | 4.1 (3.9 to 4.3) | 16018 | 12.0 (11.8 to 12.2) | 8.5 (8.4 to 8.7) |
| All sites but skin | 81167 | 336.7 (334.4 to 339.1) | 271.5 (269.6 to 273.4) | 444323 | 333 (332 to 333.9) | 230.5 (229.8 to 231.2) |
| All sites | 81709 | 339.0 (336.7 to 341.3) | 273.4 (271.5 to 275.2) | 449934 | 337.2 (336.2 to 338.1) | 233.4 (232.7 to 234.1) |

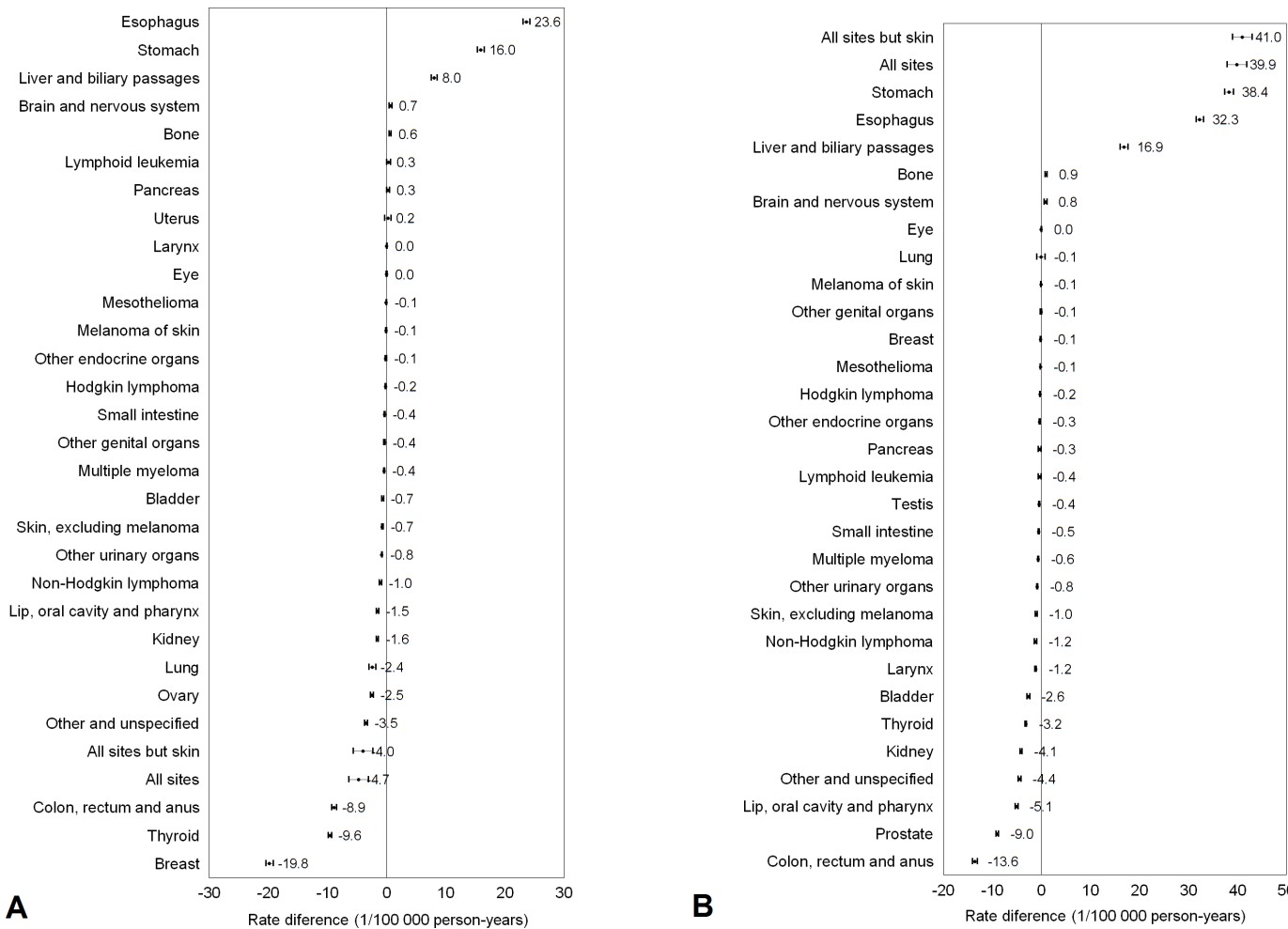

**Figure 1** Rate differences and 95% CIs in cancer incidence comparing women (A) and men (B) in rural versus urban areas in China, 2008–2012.

breast cancer. Men in rural areas had higher incidence rates of cancers of the oesophagus and stomach, but 50% or lower incidence rates of cancers of the thyroid, kidney, prostate, mesothelioma, testis, Hodgkin lymphoma, lip, oral cavity and pharynx, and small intestine.

## DISCUSSION

The present study revealed substantial disparities in cancer incidence between rural and urban areas in China, and the patterns varied across cancer types and the sexes. The overall cancer incidence was higher in men living in rural areas compared with those in urban areas but showed less urban–rural disparity in women. Both men and women in rural areas had substantially higher incidence rates of cancers of the oesophagus, stomach, and liver and biliary passages, while higher incidence rates were observed in urban areas for a number of other cancers, including cancers of colon and rectum, thyroid, breast, prostate, lip, oral cavity and pharynx, and kidney. This study provided an updated assessment of the urban–rural disparity in cancer incidence in China on a nationwide scale.

Overall, our findings were consistent with previous reports. A study using data from 72 registries in the

Chinese National Central Cancer Registry in 2009 showed that oesophageal cancer was one of the most common cancers in China, and the incidence rate was particularly high in men in rural areas.[3] A more recent study including 177 cancer registries further confirmed the higher incidence and mortality rates of oesophageal cancer in rural areas than in urban areas in both sexes 2011.[4] In this study, we found that the incidence rate of oesophageal cancer in women was 9.8 times higher in rural areas than in urban areas, and men in rural areas also had 3.9 times higher risk of oesophageal cancer than those in urban areas. Such striking urban–rural disparity may be partially explained the higher prevalence of major risk factors for oesophageal cancer, including lower socio-economic status, tobacco smoking, heavy alcohol use and dietary factors, in rural areas.[14 15] On the other hand, the aetiology of oesophageal cancer, particularly that of the main histological subtype oesophageal squamous cell carcinoma in China, has not been fully elucidated.[16] There may be other mechanisms explaining the burden gap between rural and urban populations in China.

Higher incidence rates of stomach and liver cancers in rural areas than in urban areas in China have also been

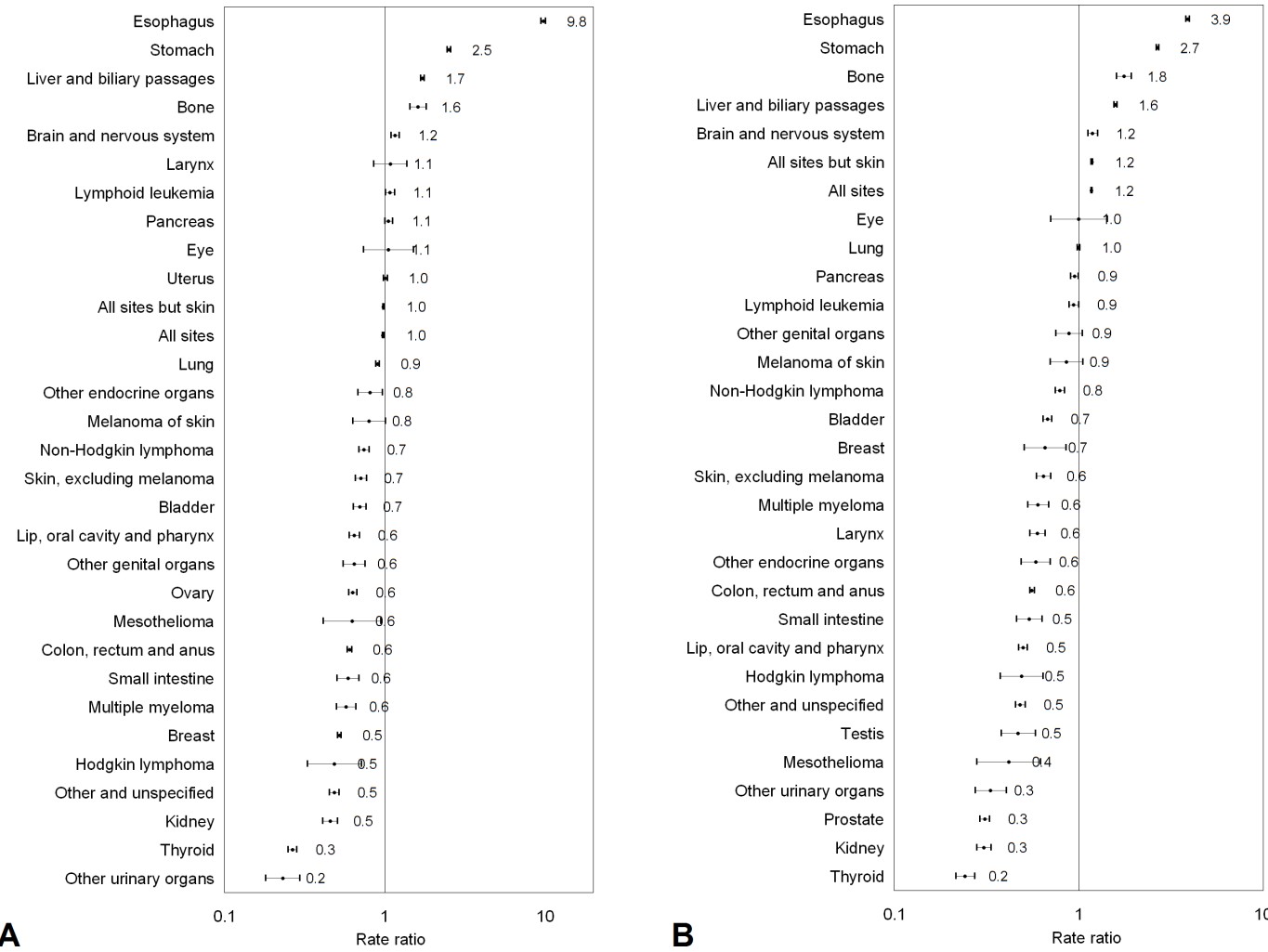

**Figure 2** Rate ratios and 95% CIs in cancer incidence comparing women (A) and men (B) in rural versus urban areas in China, 2008–2012.

reported previously.[5 6] Our findings were in line with these reports, which stressed the need for targeted prevention strategies for these cancers in rural areas. As many modifiable factors, including obesity,[17] nutrition,[18] Helicobacter pylori infection,[19] virus infection,[20] alcohol use[21] and smoking[20] are important risk factors for stomach and/or liver cancers, primary prevention strategy targeting at these dimensions may be implemented in rural areas to narrow the urban–rural disparity in these cancers.

Thyroid and breast cancers are the most common cancer types in women, especially in the middle-aged group.[22] Higher incidence rates of thyroid and breast cancers in women in urban areas than in rural areas in China have been previously reported,[7 8] which were in in line with our findings. Such urban–rural disparity might be contributable to the possible differences in prevalence of risk factors, for example, smoking, obesity, radiation exposure, oral contraceptives usage and intake of vegetables,[23–25] between women in rural and urban areas. Another potential explanation is the more frequent use of screening diagnostic procedures, such as ultrasound, Doppler examination, CT and MRI scanning, and

biochemical markers, in urban residents, which might have increased the detection of these cancers at an early stage, and thus, led to somewhat overestimated incidence of these cancers in recent years.[26] The more frequent use of screening or diagnostic procedures, such as prostate-specific antigen testing, in urban residents may also partially explain the higher incidence rate of prostate cancer in men living in urban areas than in rural areas.

We found higher incidence rates of colorectal cancer in both men and women living in urban areas compared with their counterparts in rural areas, which were in agreement with some other studies.[5 27 28] Such disparity might be explained by higher prevalence of certain risk factors, particularly those associated with Western lifestyles including low intake of fruit and vegetables, read meat consumption and lack of physical activity, in those in urban areas. However, it was also likely due to differences in access to diagnostic and treatment services between urban and rural residents. Nevertheless, considering the huge disease burden and possibility of early detection, both primary and secondary prevention strategies are highly needed for targeting urban populations, not only for minimising the urban–rural

disparity but also for reducing deaths from this cancer. A tool tick, comprehensively including lifestyle, obesity and cardiometabolic factors may be used for prevention of colorectal cancer.[29]

The aetiology of some other cancers with notable urban–rural disparity, including oral cancer and kidney cancer, has not been well illustrated. A better management regarding tobacco smoking, alcohol use, sexually transmitted disease, dental hygiene, nutrition and occupational hazards in urban residents might facilitate the reduction of disparity in the incidence of these cancers.[30 31]

There are strengths of the present study. One major strength was that we used data from the 36 cancer registers of high quality in terms of completeness of coverage and accuracy, thereby lending validity to the findings. In addition, we comprehensively investigated urban–rural disparity for overall and site-specific cancer incidence by sex and on both absolute and relative scales.

This study also has some limitations. This study was embedded in the cross-sectional design, which might not conclude the causation between geographical disparities to cancer incidence. In addition, we used data constructed at the city level. Thus, whether our findings could be generalised to the individuals needs verification. The included registries only represented a limited proportion of the total population in China and were not in well balanced distribution regarding geographical regions, and thus, the strengths of the disparity measures may not exactly reflect the extent of nationwide urban–rural disparity in cancer incidence. Due to unavailability of complete data in all cancer registries, we were not able to analyse the data by histological type, or the associations of gross domestic product or human development index with ASRs. Last but not least, the analyses were based on cancer incidence data only without information on risk factors, and we could not direct evaluate how the observed disparity could be explained by the corresponding risk factors.

## Conclusions

This updated assessment of the urban–rural disparity in cancer incidence in China revealed substantial urban–rural disparity which varies across cancer types and the sexes. The observed urban–rural disparity may be explained by a combination of differential prevalence of risk factors and access to screening and diagnostic services. Cancer prevention strategies should be tailored for common cancers in rural and urban areas.

**Acknowledgements** We would like to thank International Agency for Research on Cancer for sharing data.

**Contributors** S-HX designed the study, analysed the data and reviewed the article. SY analysed the data and drafted and reviewed the article. All authors read and approved the final manuscript.

**Funding** This study was supported by the Swedish Cancer Society (grant number 190043) and Karolinska Institutet Research Foundation (grant number 2018-01558).

**Competing interests** None declared.

**Patient consent for publication** Not required.

**Ethics approval** The analyses were solely based on publicly available data of population sizes and aggregated number of cancer cases and as such, ethics approval or consent to participate was not deemed to be necessary.

**Provenance and peer review** Not commissioned; externally peer reviewed.

**Data availability statement** Data are available in a public, open access repository. All data used are publicly available from the International Agency for Research on Cancer (IARC).

**ORCID iD**
Shuai Yuan http://orcid.org/0000-0001-5055-5627

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
