## [Reviewer comments · BMJ Open]

ARTICLE DETAILS

TITLE (PROVISIONAL)	Urban-rural disparity in cancer incidence in China, 2008-2012: A cross-sectional analysis of data from 36 cancer registers
AUTHORS	Yuan, Shuai; Xie, Shao-Hua

VERSION 1 – REVIEW

REVIEWER	Wanqing Chen National Cancer Center/National Clinical Research Center for Cancer/Cancer Hospital, Chinese Academy of Medical Sciences and Peking Union Medical College, Beijing, China
REVIEW RETURNED	19-Aug-2020

GENERAL COMMENTS	This study aimed to quantify the urban-rural disparity in cancer incidence in China. The authors used data from the 36 cancer registers which are included in the XI volume of Cancer Incidence in Five Continents. This study concluded with a substantial urban-rural disparity in cancer incidence in China. The topic is important. But the urban-rural disparity in cancer burden has been described well in China and seems to be axiomatic (Chen, Chin J Cancer Res, 2017; Yang, Chin J Cancer Res, 2017; Chen, Zhonghua Zhong Liu Za Zhi, 2014). This manuscript cannot satisfy the need for the novelty to be published. The authors' own original work accounted for a small proportion of this study. The data were extracted from the online tool by IARC, which can give both crude and age-standardized rates. The authors got the rate difference and rate ratio after simple subtractions and divisions without further analysis. As the authors stated, the 36 cancer registers covered a tiny proportion of the total population in China. This manuscript cannot align with its title "Urban-rural disparity in cancer incidence in China". In the introduction, the authors claimed that previous studies were limited in a certain area or specific types of cancer. However, the paper by Chen (Chin J Cancer Res, 2017) has described cancer patterns in areas with different urbanization rates using data from 347 cancer registries. The definition of rural and urban areas was missed in the method section. Besides, the division was too rough. As mentioned above, the variable used by Chen (Chin J Cancer Res, 2017), that is, urbanization rates could better describe the urbanization level.
---

REVIEWER	Wen Denggui 4th hospital of hebei medical university
REVIEW RETURNED	22-Sep-2020

GENERAL COMMENTS	urban-rural disparity in cancer reflect unequall socioeconomic development . Difference in ASRs may be examined in relation to GDP or HDI. Second, the authors should analyze in comparison with previous studies is the urban-rural disparity in UGIC or westernization-related cancer (colorectal or female breast cancer) increasing or decreasing ? Third, though registry-specific risk factor is unavailable, but trend in diet transition, air pollution, urbanization may be discussed in relation to the disparity.
---

REVIEWER	Zaitu, Masayoshi The University of Tokyo, Department of Public Health
REVIEW RETURNED	14-Jan-2021

GENERAL COMMENTS	General comments: This study ecologically examined urban-rural disparities in cancer incidence in China, with a dataset of 36 regional cancer registries in 2008-2012. Overall I would like to compliment the authors' ideas. Although the interesting findings might shed light and help tackle social inequalities, I have some critical concerns and comments below. Addressing these would hopefully improve the quality of this manuscript. Specific comments: #1. No DCO data were provided, while this dataset was addressed as high-quality. Please clarify the percentage of DCO in this study. Also, what percentage does this dataset cover among the total population of China? Please clarify. #2. Definitions for each cancer site were not provided. Please specify with ICD codes, probably somewhere in a Table. #3. Result section: Showing all estimates and 95% CIs are very busy and redundant in the main text. So, please consider displaying full data on Tables/Figures, but not in the main text. Since minor cancers were not focused on the discussion, please consider making these results of minor cancers concise (probably, explaining patterns are enough). #4. Should ASR be explained as "per 100,000 population", but not as per 100,000 person-years? #5, Please reconsider the way of showing 95% CIs, with minus symbols. For instance, "the breast (-19.8; 95% CI, -20.4--19.2)" is hard to follow. Showing in the way of "-20.4 to -19.2" or "-20.4, -19.2" is better. #6. The major limitations include a cross-sectional design, which may not conclude the causation between geographical disparities to cancer incidence. Ecological design is another limitation. Namely, no individual-level socioeconomic indicators (education, income, occupation) were available. #7. Page 11, last sentence, "the higher prevalence of major risk factors for esophageal cancer, including lower socioeconomic status":
---

	What is the potential biological mechanism of lower SES that might increase esophageal cancer, other than specified risks of tobacco smoking, heavy alcohol use, and dietary factors? For instance, H. pylori infection would be one candidate for stomach cancer (Cancer Med. 2019 Feb;8(2):795-813). #8. Page 12, 3rd paragraph "Another potential explanation is the more frequent use of screening diagnostic procedures, ..., which might have increased the detection of these cancers.": Is this discussion for over-diagnosis? If so, please clarify.
--	---

VERSION 1 – AUTHOR RESPONSE

Reviewer: 1

Dr. Wanqing Chen, National Office for Cancer Prevention and Control & National Central Cancer Registry, National Cancer Center/Cancer Hospital, Chinese Academy of Medical Sciences and Peking Union Medical College, Beijing

Comments to the Author:

This study aimed to quantify the urban-rural disparity in cancer incidence in China. The authors used data from the 36 cancer registers which are included in the XI volume of Cancer Incidence in Five Continents. This study concluded with a substantial urban-rural disparity in cancer incidence in China. The topic is important. But the urban-rural disparity in cancer burden has been described well in China and seems to be axiomatic (Chen, Chin J Cancer Res, 2017; Yang, Chin J Cancer Res, 2017; Chen, Zhonghua Zhong Liu Za Zhi, 2014). This manuscript cannot satisfy the need for the novelty to be published.

Response: We are aware of these previous studies reporting urban-rural disparity in cancer burden in China. However, none of these qualitatively measured such disparity, either on absolute (in terms of rate difference) or on relative scale (in terms of rate ratio), as in our study. We believe that the reviewer should have been advised not to judge importance or breadth of appeal by the Journal. The readers would be able to make these judgements for themselves. The reviewer may wish to refer to Instructions for reviewers (<https://bmjopen.bmj.com/pages/reviewerguidelines>).

The authors' own original work accounted for a small proportion of this study. The data were extracted from the online tool by IARC, which can give both crude and age-standardized rates. The authors got the rate difference and rate ratio after simple subtractions and divisions without further analysis.

Response: We disagree with such dismissal of work and possibly misunderstanding of methodologies. We extracted raw data, i.e., the sex- and age-specific numbers of cases and corresponding population sizes for the around 30 cancer types from the 36 cancer registers. We pooled the incidence rates separately for rural and urban areas by sex and cancer type and calculated the rate difference and rate ratio for each cancer by sex. Confidence intervals were also estimated for all these measures.

As the authors stated, the 36 cancer registers covered a tiny proportion of the total population in China. This manuscript cannot align with its title "Urban-rural disparity in cancer incidence in China".

Response: As discussed in the manuscript, we are aware of the limited generalizability of our results. However, this would not have affected the interval validity of our study which was based on the

selected cancer registries with high-quality data. Nevertheless, we have revised the title as “Urban-rural disparity in cancer incidence in China, 2008-2012: A cross-sectional analysis of data from 36 cancer registers”.

In the introduction, the authors claimed that previous studies were limited in a certain area or specific types of cancer. However, the paper by Chen (Chin J Cancer Res, 2017) has described cancer patterns in areas with different urbanization rates using data from 347 cancer registries.

Response: First, we stated that “most of” rather than “all” these previous studies were limited in a certain area or province, or specific types of cancer only. Second, as mentioned above, the paper of Chen 2017 did not involve quantitative assessment, i.e., in terms of rate difference or rate ratio or any other disparity measure, of the urban-rural disparity as did in the present study. Third, we would be conservative to judge a study involving 347 cancer registries as with higher validity than our analysis, because not all these 347 cancer registries have provided high quality data (otherwise all of these would have been included in the Cancer Incidence in Five Continents series). By the way, we do not think we are able to obtain data from more cancer registries beyond those publicly available in the Cancer Incidence in Five Continents series. For example, in a previous study from our research group (Wang QL, Clin Epidemiol 2018;10:717-728), we updated the global trends of incidence of esophageal squamous cell carcinoma, when we were able to obtain data from 30 cancer registries from 20 countries all over the world through 2015. However, the reviewer, who was in charge of cancer registration in China, refused to share their data (which are financed by public funding and should be accessible to researchers). It was very unfortunate to conduct the study without data from China, although for which half of global esophageal cancer cases were from China.

The definition of rural and urban areas was missed in the method section. Besides, the division was too rough. As mentioned above, the variable used by Chen (Chin J Cancer Res, 2017), that is, urbanization rates could better describe the urbanization level.

Response: The categorization of rural or urban areas is listed in Table 1. Such categorization was made referring to reports by the reviewer (e.g., Chen et al, J Thorac Dis. 2013). We do not entirely agree that urbanization rate (the proportion of non-agricultural population) is the best to characterize urbanization level, as the definition of “non-agricultural population” does not necessarily reflect the population who are actually involved in agricultural industry. The classification of agricultural and non-agricultural residency status (known as hukou system) is somewhat outdated. Moreover, considering the varying latency periods from exposure to risk factors and onset of different cancers, selection of the specific year/period of data on urbanization rates would be problematic when making comparisons across cancer types.

Reviewer: 2

Dr. Denggui Wen, 4th hospital of hebei medical university

Comments to the Author:

urban-rural disparity in cancer reflect unequal socioeconomic development. Difference in ASRs may be examined in relation to GDP or HDI.

Response: Thank you for the comment. We agree that it will be interesting to examine the association of ASRs with GDP or HDI. Unfortunately, GDP and HDI data were unavailable for all studied areas in this study. Thus, we did not examine the incidence rates in relation to GDP or HDI. Instead, we have added the following text acknowledging the limitation in the Discussion.

Page 11:

“Due to unavailability of complete data in all cancer registries, we were not able to analyze the data by histological type, or the associations of socioeconomic indicators, e.g. gross domestic product or human development index, with ASRs.”

Second, the authors should analyze in comparison with previous studies is the urban-rural disparity in UGIC or westernization-related cancer (colorectal or female breast cancer) increasing or decreasing?

Response: In the Discussion, we have compared our findings regarding these cancers with previous studies as follows: the findings from Chinese National Central Cancer Registry in 2009 (the 2nd paragraph, Page 8), the findings from 177 cancer registries (the 2nd paragraph, Page 8), and several other studies. However, considering the different study populations across studies, we did not make any inference on the time trends of the rural-urban disparity.

Third, though registry-specific risk factor is unavailable, but trend in diet transition, air pollution, urbanization may be discussed in relation to the disparity.

Response: We agree that relating the observed disparity with trends in prevalence of risk factors would be valuable. However, this study was cross-sectional based data in a 5-year period only and the change of diet, environment and urbanization might be minimal. We believe investigating the time trends in the disparity and examining their relationship with risk factors is worth a separate study.

Reviewer: 3

Dr. Masayoshi Zaitu, The University of Tokyo

Comments to the Author:

General comments:

This study ecologically examined urban-rural disparities in cancer incidence in China, with a dataset of 36 regional cancer registries in 2008-2012. Overall I would like to compliment the authors' ideas. Although the interesting findings might shed light and help tackle social inequalities, I have some critical concerns and comments below. Addressing these would hopefully improve the quality of this manuscript.

Response: Thank you for your comments.

Specific comments:

#1. No DCO data were provided, while this dataset was addressed as high-quality. Please clarify the percentage of DCO in this study. Also, what percentage does this dataset cover among the total population of China? Please clarify.

Response: We have now provided DCO data for each registry in the Supplementary Table 2. The DCO value ranged from 0 to 11.8. The estimated case-weighted mean of DCO was ~1.9 in male, ~1.5 in female and ~1.7 in both male and female among included registries. The 36 cancer registries together covered approximately 5% of the total population in China. We have added the following text in the Methods section.

Page 5:

“The case-weighted means of percentage of case defined by death certificate only were ~1.9 in male, ~1.5 in female and ~1.7 in both male and female among included registries. Included population from 36 cancer registries made up ~5% of the total population in China”

#2. Definitions for each cancer site were not provided. Please specify with ICD codes, probably somewhere in a Table.

Response: The ICD codes for all studied cancers are provided by IARC in its webpage as the PDF format (<https://ci5.iarc.fr/CI5-XI/PDF/Chapter%203.pdf>). We have added this information in our manuscript.

#3. Result section: Showing all estimates and 95% CIs are very busy and redundant in the main text. So, please consider displaying full data on Tables/Figures, but not in the main text. Since minor cancers were not focused on the discussion, please consider making these results of minor cancers concise (probably, explaining patterns are enough).

Response: Thank you for the comment. We have now removed estimates and corresponding 95% CIs from the main text.

#4. Should ASR be explained as "per 100,000 population", but not as per 100,000 person-years? Could be "per 100,000 population"

Response: The unit has been revised to per 100,000 population per year as suggested.

#5, Please reconsider the way of showing 95% CIs, with minus symbols. For instance, "the breast (-19.8; 95% CI, -20.4--19.2)" is hard to follow. Showing in the way of "-20.4 to -19.2" or "-20.4, -19.2" is better.

Response: Revised as suggested.

#6. The major limitations include a cross-sectional design, which may not conclude the causation between geographical disparities to cancer incidence. Ecological design is another limitation. Namely, no individual-level socioeconomic indicators (education, income, occupation) were available.

Response: We agree that both cross-sectional and ecological designs of the present study are limitations. We have now acknowledged in the Discussion part.

Page 11:

"This study was embedded in the cross-sectional design, which might not conclude the causation between geographical disparities to cancer incidence. In addition, we used data constructed at the city-level. Thus, whether our findings could be generalized to the individuals needs verification."

#7. Page 11, last sentence, "the higher prevalence of major risk factors for esophageal cancer, including lower socioeconomic status": What is the potential biological mechanism of lower SES that might increase esophageal cancer, other than specified risks of tobacco smoking, heavy alcohol use, and dietary factors? For instance, H. pylori infection would be one candidate for stomach cancer (Cancer Med. 2019 Feb;8(2):795-813).

Response: SES is an upstream health determinant to cancer via its influence on health behaviors, such as smoking, drinking, diet, and other factors, such as virus infection, stress, working environment, etc. In addition, SES differs largely between urban and rural areas in China. SES, therefore, might be an important driver for the geographic cancer disparity in China. We agree with the reviewer and admit that SES derived cancer disparity is a complicated phenomenon.

We have also added Helicobacter pylori infection as a factor related to SES and the suggested reference in the manuscript

Page 9

“including obesity 17, nutrition 18, Helicobacter pylori infection 19, virus infection 20, alcohol use 21 and smoking 20 are important risk factors for stomach and/or liver cancers,”

#8. Page 12, 3rd paragraph "Another potential explanation is the more frequent use of screening diagnostic procedures, ..., which might have increased the detection of these cancers.": Is this discussion for over-diagnosis? If so, please clarify.

Response: Yes, we meant that the better health care may detect certain types of cancer at the earlier stage in urban residents compared to rural residents in recent years, which might have cause artificially overestimated higher incidences of corresponding cancers in urban areas during the study period. We have revised the text as follows.

Page 10:

“Another potential explanation is the more frequent use of screening diagnostic procedures, such as ultrasound, Doppler examination, CT and MRI scanning, and biochemical markers, in urban residents, which might have increased the detection of these cancers at an early stage, and thus, led to somewhat overestimated incidence of these cancers in recent years”.

VERSION 2 – REVIEW

REVIEWER	Chen, Wanqing National Office for Cancer Prevention and Control & National Central Cancer Registry, National Cancer Center/Cancer Hospital, Chinese Academy of Medical Sciences and Peking Union Medical College, Beijing
REVIEW RETURNED	01-Feb-2021

GENERAL COMMENTS	Although I could not agree to all responses from authors about comments raised, I still support this manuscript would be published. Additionally, I would like to make cancer registry data available but I can't without permission by budget providers. Moreover, we did not submit our data completely to CI5, and some registries not met the criteria of CI5 did not mean the quality was poor due to different requirement and rules. We did quality control very district and about 1/3 of registries data are declined every year by annual report.
---

REVIEWER	Zaitu, Masayoshi The University of Tokyo, Department of Public Health
REVIEW RETURNED	28-Jan-2021

GENERAL COMMENTS	I thank the authors. All concerns raised in the previous version have been revised appropriately—no further comments.
---

VERSION 2 – AUTHOR RESPONSE

Reviewer: 1

Dr. Wanqing Chen, National Office for Cancer Prevention and Control & National Central Cancer Registry, National Cancer Center/Cancer Hospital, Chinese Academy of Medical Sciences and Peking Union Medical College, Beijing

Comments to the Author:

Although I could not agree to all responses from authors about comments raised, I still support this manuscript would be published. Additionally, I would like to make cancer registry data available but I

can't without permission by budget providers. Moreover, we did not submit our data completely to CI5, and some registries not met the criteria of CI5 did not mean the quality was poor due to different requirement and rules. We did quality control very district and about 1/3 of registries data are declined every year by annual report.

Response: Thank you for reviewing our paper. We appreciate your offer of data to CI5.

Reviewer: 3

Dr. Masayoshi Zaitu, The University of Tokyo

Comments to the Author:

I thank the authors. All concerns raised in the previous version have been revised appropriately—no further comments.

Response: Thank you for reviewing our paper.